# The Lived Experiences and Perspectives of People with Autism Spectrum Disorder in Mainstream Employment in Australia

Melissa Sharpe [1], Claire Hutchinson [1,2,*] and June Alexander [1,*]

1   College of Nursing and Health Sciences, Flinders University, GPO Box 2100, Adelaide, SA 5001, Australia; melsharpe28@hotmail.com
2   Caring Futures Institute, Flinders University, GPO Box 2100, Adelaide, SA 5001, Australia
*   Correspondence: claire.hutchinson@flinders.edu.au (C.H.); june.alexander@flinders.edu.au (J.A.)

**Abstract:** Individuals with autism spectrum disorder (ASD) experience significant barriers to employment. This study aimed to look at the support received by individuals with ASD in gaining and maintaining open employment from their perspective. A phenomenological approach was adopted with participants (n = 9) participating in semi-structured interviews. Thematic analysis identified four key themes; being supported, feeling successful, career progression and expectations. The findings suggest that individuals with ASD receive support from many different sources in their employment including supervisors, co-workers and parents. The support they received from disability employment consultants was more focused on obtaining a job rather than job maintenance or career progression. Career progression was rarely discussed by participants with their disability employment consultant, despite some evidence of poor job matches and unused qualifications and skills (reflecting a poor investment for individuals and society). Despite this, participants reported feeling successful due to having a job, having pride in their work, and feeling valued by co-workers. The study highlights the need for more research on understanding the longer-term support needs of people with ASD in open employment. Changes in policy to better resource and incentivize disability employment providers could produce more positive outcomes for people with ASD throughout their working lives.

**Keywords:** autism spectrum disorder; disability employment services; job matching; open employment; support needs

## 1. Introduction

Diagnosis of autism spectrum disorder (ASD) worldwide has increased in recent decades [1]. ASD is categorised as a developmental disability that can impact social skills and communication and is often typified by repetitive patterns of behaviour and intense interests or activities [2]. Though assessments vary by country, resulting in significant differences in the reporting of the prevalence of ASD across the world [1], the prevalence in Australia is estimated to be about 2.5% [3]. The social challenges experienced by people with ASD often intensify in adolescence when relationships can become more complex for young people to navigate [4]. Consequently, people with ASD face many barriers when they seek employment, such as difficulties with the job interview process, communication and understanding workplace culture [5–9]. Despite people with ASD often having high-level or multiple qualifications [7,8,10], they experience lower levels of employment than the general population. In Australia, only 38% of adults with ASD are employed compared to 84.1% of Australians of working age without disability [11], and people with ASD are less likely to be employed than people with intellectual disability [5,12].

Similar patterns of low employment for people with ASD have been observed in other developed countries [13,14]. However, individuals with ASD report that working is important to them [15], and research has highlighted a number of positive outcomes from

gaining and maintaining employment including skills development [16], better mental health [17], development of social relationships [16,18], better life satisfaction [19], well-being [13], and better quality of life [20]. In the general employment literature, even part-time work for as little as eight hours a week has been shown to result in better mental health and positive well-being outcomes [21]. Furthermore, studies of employees with ASD have identified a range of organisational benefits from their employment including close attention to detail, good long-term memory and ability to concentrate, as well as being reliable and having lower levels of absence [5,22].

Research to identify facilitators and barriers to the employment of people with ASD from the perspectives of people with ASD, their family members, employers, service providers and researchers across Australia, Sweden and the US identified that a strengths-focused approach and good quality job matching were the most important facilitators to successful employment, whilst stigma, communication challenges and a lack of understanding about ASD were considered to be key barriers [23]. However, disability employment consultants are not always trained or equipped to support people with ASD and do not necessarily take a strengths-based approach [5,24]. Furthermore, employers may not be well placed to assess the skills and capabilities of people with ASD during recruitment. A recent international study identified that being asked to complete a task instead of undertaking a formal interview may be a more supportive recruitment practice for people with ASD [25].

Traditionally people with disabilities, including ASD, have worked in sheltered workshops with others with disabilities; these are known as Disability Enterprises in Australia [26]. This type of employment has been criticised as further cementing the segregation of people with disabilities, as well as offering repetitive tasks and low financial rewards which are designed to "top up" rather than replace disability pensions [26,27]. Social enterprises set up to provide employment for people with disabilities have the advantage of offering market-level wages, but research has identified that few positions exist, and people with disabilities often volunteer at such organisations whilst waiting for paid positions to become available [27]. A third employment pathway is self-employment or microenterprise; defined in Australia as small businesses employing four people or less [28]. However, whilst many businesses can be successful—especially those run by people with acquired physical disabilities and previous high-levels skills and education—the picture is more mixed for people with other types of disabilities, who may experience significant psychosocial outcomes, but are often still financially dependent on disability pensions [29,30]. A fourth pathway is that of mainstream employment or "open employment"—known as integrative or supported employment internationally [31]. Open employment involves working with co-workers with and without disability for market-level wages. Competitive integrated employment has in recent decades been prioritised because of improved outcomes when compared to sheltered employment [32,33]. In Australia, disability employment services (DES) by not-for-profit and for-profit providers are funded by the Australian Government to provide support to help people with disabilities find and maintain employment.

### 1.1. People with ASD in Open Employment

Individuals with ASD face barriers to both gaining and maintaining open employment [7,34,35], yet there is relatively little literature on the open employment of people with ASD [25]. A recent study by Harvery et al. [10] using baseline data from the Australian Longitudinal Study of Autism in Adulthood (n = 149), identified that most employed respondents were in open employment but received no support; only 6% reported receiving support to maintain their employment [10]. Furthermore, though most had tertiary level qualifications, many were employed in part-time, low-skill, entry-level positions with only a quarter receiving appropriate adjustments [10]. Most notably those experiencing higher levels of autistic traits were more likely to be employed in positions that reflected their preferences and qualifications; thereby suggesting that employers can value the unique traits of employees with ASD, such as intense focus and attention to detail [10].

Research on open employment for people with ASD has focused on both the demand and supply sides of the employment equation. That is, on employers' perspectives of employing people with ASD (demand), and strategies for supporting entry to employment and the teaching of job-related skills (supply). There has been notably less emphasis on people with ASD's own perspectives of employment and how to support the employment of people with ASD over their working lives.

Firstly, addressing the demand side of the employment equation, employers who have not previously employed people with ASD have expressed fear of the unknown when considering such employment [36]. Employers who have previously employed people with ASD have highlighted that managers and co-workers need to know how best to interact with and support people with ASD [36,37], and that external support is needed in order to achieve successful employment outcomes [36]. Support for people with ASD in the workplace has been identified as being enhanced when there is disability and ASD inclusion training [36,37]. Lack of understanding from co-workers and supervisors and a lack of appropriate adjustments can represent barriers to the employment of people with ASD [22,24,38].

Employers may be concerned about additional accommodations and costs associated with the employment of people with ASD, though research suggests that the cost of accommodations for people with ASD are relatively minor, especially compared to the accommodations required to support the employment of those with physical or sensory disability [39]. An employer survey study conducted in Australia found that employing a person with ASD does not cost any more than employing any other employee, and that employees with ASD displayed not only strong attention to detail but also a good work ethic [40].

On the supply side of the employment equation, the readiness of people with ASD to gain open employment has been enhanced via video modelling, role play and group training in social and communications skills, as well as job-specific vocational skills [8,41,42]. A recent scoping review of disability employment interventions for people with ASD identified 36 studies that evaluated employment interventions [8]. The authors noted two prominent trends in these studies. Firstly, that studies were primarily focused on improving the performance of people with ASD, and essentially ignored the impact of organisational barriers on employment outcomes. Secondly, most of the studies focused on gaining employment, with only eight included studies focused on maintenance of employment [8]. Therefore, there is little evidence relating to long-term support to maintain employment nor any detailed exploration of the perspectives of employees with ASD themselves.

*1.2. The Current Study*

This study seeks to address a gap in the literature relating to the perceptions of individuals with ASD on the support they receive to gain and maintain employment from their own lived experiences. This study adopts a phenomenological approach using qualitative methodology to address the following research aims:

1. To gain perspectives of people with ASD about which supports they find most effective in gaining and maintaining employment.
2. To determine how success in employment is measured by employees with ASD themselves.

## 2. Methods

This study was based on a social constructivist paradigm and used a phenomenological qualitative approach. This approach allows for the exploration of—and the drawing of meaning from—the perspectives and lived experiences of participants [43,44]. As such, the focus is on describing individuals' conscious experience without seeking to identify casual relationships; seeking to identify convergence of experience as well as divergence [45,46]. Given these aims, one-to-one semi-structured interviews were selected as the most appropriate methods to explore the perceptions of individuals with ASD and to generate rich

descriptions of their lived experiences [47]. Ethics was granted through the Flinders University Social and Behavioural Research Ethics Committee (Approval number: SBREC 8615).

## 2.1. Recruitment

Three disability organisations from metropolitan Adelaide, South Australia supported the recruitment of participants. An overview of the project, participant information sheets and consent forms were provided to the organisations to distribute to clients that met the criteria for the study. To participate in the study, participants needed to have a diagnosis of ASD (self-report), be over the age of 18, speak English, and be employed in open employment. Participants gave formal written consent to participate and for organisations to provide their contact details to the first author to further discuss the study and organise a suitable time for the interview.

## 2.2. Participants

This study included nine individuals with self-reported ASD who were employed in open employment. Although three providers agreed to support recruitment, all nine participants received support from the same provider. Participants were aged between 22 and 32 years old (mean = 22.5 years old). There were seven male and two female participants with all but one having held some form of employment previously. Participants had been employed in their current workplace for between two months and eight years (mean = 3.5 years). Most participants had several qualifications and reported their current employment as entry-level (Table 1).

## 2.3. Materials

Participants took part in one semi-structured interview. This was conducted by the first author using a semi-structured interview guide to ensure the interview was on-topic whilst also allowing for broader perceptions to be explored based on topics raised by the participant [48]. Examples of open questions asked are: what have you found most helpful in getting a job? Why was that helpful? What have you found most helpful in keeping a job? Is there any other support you would have liked when you were looking for a job? Is there any other support you would like now? What are the benefits of having a job for you? Participants were also asked some basic socio-demographic questions and about their education and employment history. On average the interviews lasted 26 and a half minutes.

## 2.4. Procedure

Interviews were conducted at a place of the participants choosing with participants able to be accompanied by a person for support and advocacy if they wished. Two participants selected this option, which in both cases was the participants' key worker. On a few occasions, the key worker helped rephrase questions to support participants' understanding but otherwise were not involved in the interview. Interviews were audio-recorded with participants' consent, transcribed by the first author, and de-identified prior to analysis.

## 2.5. Data Analysis

Thematic analysis was used to analyse the data collected through the interviews supported by NVivo 12 software (QSR International Pty Ltd., Melbourne, VIC, Australia). The thematic analysis allowed the researchers to sort the data into codes and then into broader themes whilst retaining the voice of the individuals with ASD [45,49]. The coding was conducted in two stages. In the first stage, the first author created a set of initial codes based on key topics discussed during the interviews [45], noting that data saturation was achieved with no new codes emerging by the time the final transcript was coded [50]. At the second stage, these initial codes were reviewed by the authorship team and, following discussion and reference to the research aims, were collated into broader themes. These

themes aimed to represent the lived experiences of the participants and capture divergence, as well as convergence, of experiences [45,46].

**Table 1.** Participant information.

| | Age Range | Gender | Employment Type | Role | Length of Employment | Previous Employment | Qualifications [1] |
|---|---|---|---|---|---|---|---|
| 1 | 25–29 | Female | Retail | Back of house, food preparation | 6–12 months | • Dog groomer<br>• Retail/admin | Cert 1 & 2 in animal studies<br>Cert 1 & 2 in digital media |
| 2 | 20–24 | Male | Hospitality | Cleaning carpark, restaurant and occasionally serving people, food service | 3–5 years | • n/a | TAFE course in retail (level not specified) |
| 3 | 20–24 | Male | Cabinet Making | Making parts for kitchen bench work | Less than 6 months | • Bartender<br>• Retail assistant | Cert 3 cabinet making and joinery<br>Responsible serving of alcohol (RSA) |
| 4 | 20–24 | Male | Retail | Cutting and preparing food items for sale | Less than 6 months | • Construction<br>• Charity work<br>• Factory work | RSA<br>Cert 1 in tourism |
| 5 | 25–29 | Male | Cleaning | Cleaning offices and office buildings | 6–10 years | • Hospitality | Cert 2 in kitchen operations<br>Cert 1 in front of house |
| 6 | 25–29 | Male | Food Service | Putting together food orders | 1–2 years | • Retail | Cert 3 in Investigative studies |
| 7 | 25–29 | Male | Hospitality | Back of house, preparing and cooking orders | 6–10 years | • Butcher | Cert 2 & 3 in hospitality<br>Barista course |
| 8 | 20–24 | Female | Animal Hospital | Assisting vet nurse and general care of animals | 3–5 years | • Animal hospital | Cert 2 & 3 in animal studies |
| 9 | 30–34 | Male | Cleaning | Cleaning offices and office buildings | 3–5 years | • Geo scientist<br>• Food service | Diploma in geo-science |

Note. [1] A note about qualifications and international equivalency. TAFE (Technical and Further Education) colleges in Australia offer vocational qualifications and are similar to community colleges in the US. TAFE qualifications are Certificates 1, 2, 3 and 4, diploma and advanced diploma. Certificates 3 and 4 are equivalent to Year/Grade 12 or GED (Graduation Equivalency Diploma) in the US. Certificates 1 and 2 are the equivalent of Year/Grade 10 and 11, Diploma is equivalent of first-year bachelor's degree, Advanced Diploma is equivalent to second-year bachelor's degree or an Associate Degree in the US.

## 3. Results

Thematic analysis identified four key themes: being supported, feeling successful, lack of career progression, and expectations, as well as seven sub-themes (Figure 1).

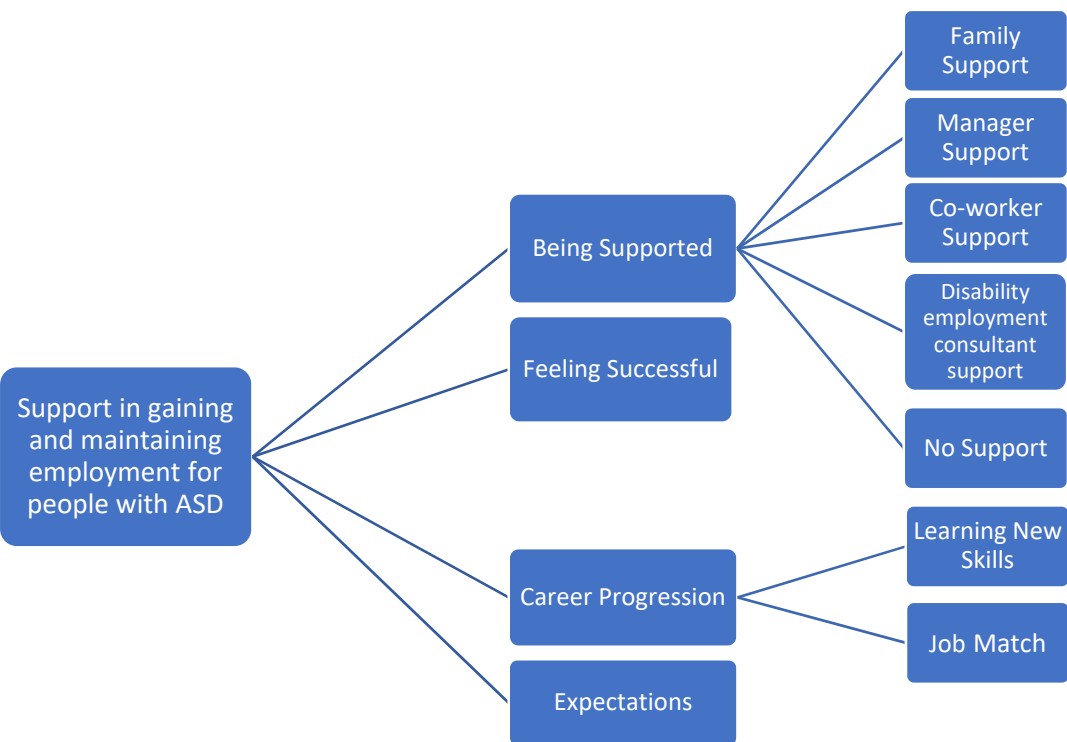

**Figure 1.** Themes and sub-themes.

### 3.1. Being Supported

A commonly discussed topic by participants was the various sources from which they received support in gaining and maintaining employment. This theme is broken into sub-themes, these were: family, manager, co-worker and disability employment consultant support. Participants also identified ways in which they received little or no support.

Most participants reported that their family supported them in both gaining and maintaining employment. Some participants reported that they obtained their job due to their parents' connections. For example, "*Ahh, Basically, ahh, mainly my mum just found the job, my mum just found the job for me [sic]*" [Participant 6]. Participants also discussed the excitement and emotional support from their families had when they found out the participant had a job: "*They were so excited when they even heard about a trial shift*" [Participant 1].

Once participants obtained a job, they commonly reported receiving support from their supervisors and/or managers. In the following quotes, the participants identify that their managers knew them well, understood their strengths as well as their weaknesses, and supported them in developing their skills.

> *[Manager's name] which is the lady that gave me the job, she's really understanding with my weaknesses and strengths.* [Participant 8]

> "... *If I had any problems, like for example, maybe I need to kind of, be shown again how to do something, because maybe I wasn't confident in doing it. I can ask my supervisor, or maybe someone that has that been [sic] in the job for a long time, just kind of show me like be like OK this is how you do it. And then they show me, and then they would see if I can do it as well*" [Participant 5]

In the following quote, the manager was clear with the participant about the ultimate goal of his training and the range of skills he would need to master: "*yeah the um eventual eventuality [sic] that my boss wants me to build a kitchen all by myself*" [Participant 3].

As well as managers and supervisors, participants also reported they received support from co-workers in learning job skills. In some cases, participants were able to build relationships with co-workers which was important to them.

*So, I've made a friend at work. We catch up for lunch every second week. Just for lunch and another friend she used to work with we catch up . . . We catch up, we socialize, we talk to each other. We hang out. I made a friend.* [Participant 1]

When it came to the support from disability employment consultants, most participants reported receiving support in writing their resume: *"Yes, they actually helped me overhaul my resume"* [Participant 3]. However, once employed, the majority of participants reported that their disability employment consultant primary support consisted of 'checking in'. Two participants reported that this was performed away from their workplace as their employer had requested the consultants do not come onto the worksite.

*With [workplace name] it's it's [sic] a bit harder because they have they have they have [sic]. They haven't requested him coming on to the site. I think. So basically, from what I've heard from [disability employment consultant name] is that he does it over the phone, but with the managers to keep up.* [Participant 6]

Two participants reported receiving no support in employment. Participant 8 reported that she did not have a DES provider when looking for a job but, once employed, she obtained support from a DES provider to support the maintenance of her employment: *"So to begin with, I didn't have any support. So yeah, I just went in blind from Dad saying, yeah, you have a chance to work here"* [Participant 8].

There was also evidence of poor job matching, for example, Participant 9 reported he had been put in a job he was not experienced enough to maintain by a previous job provider.

*. . . Problem was, I was taking too long to do it, 'cause they literally the way that I was set out to be they said they thought [sic] I had had like a few years' experience, but I hadn't, and I was hoping to be like at the lower level. But what they were trying to do is they're trying to get me to take over someone else's job* [Participant 9]

### 3.2. Feeling Successful

Participants reported feeling successful in their employment. This was recounted to be the result of staying employed, experiencing good mental health, and having pride in their work. Participant 8 talked about having pride in her work and her workplace, and that she felt confident she was a key part of the work team.

*"Uh, just that they are quite reliant on me being able to help the volunteers and stuff . . . And that they are confident that I know what I'm doing . . . And that I'm really good at what I do as well"* [Participant 8]

Participant 9 joked, *"Well, the fact is that I haven't been sacked yet"* [Participant 9] as an indicator of his success. Participant 3 talked about having a good quality of life due in large part to his employment: *"I'm happy, I'm genuinely happy and at the end of the night like end of the day, I can go to bed and actually go to sleep easily"* [Participant 3].

Financial stability was another aspect of feeling successful that was discussed by multiple participants which often resulted in them feeling more independent. In the case of Participant 8, her employment had led to her being able to move away from the family home and live independently: *" . . . feeling more independent within myself, 'cause if I didn't have the job, I wouldn't be able to be living by myself, which I am now"* [Participant 8].

### 3.3. Career Progression

Career progression was identified as a key theme, with two subthemes: learning new skills and job match. Learning new skills was commonly reported by participants. Some of these new skills were requirements of the role and crucial to maintaining employment. For example, *"Uh, with the forklift that's been, that's been, that's been [sic] a recent thing 'cause I think I needed that because they they [sic] made some upgrades"* [Participant 6].

However, four participants reported future career aspirations that did not match their current role. For example, Participant 2 reported that he wanted to work in the transport industry but was currently working in the fast-food sector. He reported that he had just

gone where his disability employment provider sent him: "*Well, whatever that whatever that came up for them they sent to me*" [Participant 2].

It was identified that five participants had qualifications in areas in which they were not currently working. An example of this is Participant 1, who has qualifications in animal studies but is working at a supermarket, and participant 4 has qualifications in tourism but is currently working at a fruit and vegetable shop (Table 1). This evidence suggested poor overall job matching with a focus on getting a job per se rather than getting a job that might be more appropriate in the long term and match participants' preferences, knowledge, skills and aspirations.

### 3.4. Expectations

The general expectation reported by participants was that the DES provider they used was doing the best job they could to support them. "*I mean I think I think they did as best as best they could. Just like finding jobs online and all and all that stuff*" [Participant 6].

Despite this, Participant 4 reported that he felt a more formal meeting was required to discuss his future career aspirations with his disability employment consultant and that the usual 'checking in' session was not the best time to raise such a discussion: "*I feel like it had to be more of when we have a sit down meeting as opposed to him coming in and being like hey how you doing, right?*" [Participant 4].

Although participants raised future aspirations, they sometimes lacked confidence in their ability to achieve different future work opportunities. For example, Participant 2 reported feeling that he may not have the skills or abilities to meet his career aspirations, and this held him back from initiating a conversation with his disability employment consultant; "*Hopefully if I talk to her about it, or even if. Or even if I have enough. Enough of my abilities to get towards that goal then yeah*" [Participant 2].

## 4. Discussion

This study sought to illuminate the lived experiences of people with ASD working in open employment and to examine how they were supported to obtain and maintain their employment. Four main themes emerged from the thematic analysis of the data; being supported, feeling successful, lack of career progression, and expectations. Support is crucial to the gaining and maintenance of open employment for individuals with ASD and we identified that people with ASD receive support to gain and maintain their employment from various sources. In alignment with previous studies, family support was reported as being important [13,51–53]. People with ASD receive support from their families in a variety of ways, such as emotional support, advocacy and navigating services [51,54]. Whilst family support is important in navigating services and advocacy, it is also vital that the people working around an individual with ASD such as supervisors, managers and co-workers are supportive and inclusive. Notably, in this study, two participants received more direct support, accessing their employment opportunities through their parents' workplace or personal contacts.

Positive working relationships with managers and supervisors are crucial to the maintenance of employment [36]. In this current study, supportive and positive relationships were identified by multiple participants, not just with supervisors and managers but also with co-workers. It was evident that many of the participants felt valued and supported by their co-workers. As in other studies, co-worker support in learning new on-the-job skills and building relationships was indicated as particularly important to participants [36].

Disability employment consultant support is often more focused on getting an individual a job rather than maintaining employment, in both this study and existing literature [10,53,54]. All participants in this study reported 'checking in' as being the main support provided once employed. Though some participants were placed in jobs that did not reflect their skills, preferences, or future aspirations, they were largely reluctant to discuss this with their consultants and instead focused on the success they felt from just being employed. Some participants intended to pursue further discussions about their

future employment, as it seemed that they did not expect such conversations to be initiated by the consultants.

Other international studies have highlighted that disability employment consultants do not receive sufficient training to support people with ASD, and lack time given high caseloads [5,23]. In a study by McDowell et al. [55], disability employment consultants reported feeling that they are more focused on "checking boxes" than having an understanding of clients' strengths and weaknesses due to time and funding restraints. In Australia, there is currently no consistent model of support or qualification requirement for disability employment consultants [15], and studies in both Australia and the US have found that disability employment practice is often not evidence-based [15,56,57]. As in other developed nations, DES providers would benefit from upskilling and a minimum qualification level being set for employment as a disability employment consultant.

Poor job matches were evident from the current study. Literature suggests that a good job match is crucial in the maintenance of employment [36]. The understanding of a person with a disability's strengths, interests and needs is essential in this job match and job maintenance [58]. Research shows if an employment consultant understands a person's strengths, interests, and needs this assists the person with a disability to be able to set and work towards career goals which is important for career progression [58]. If the job match is poor, then the individual with ASD may find the job challenging and feel defeated [36]. Wehman et al. [33] reported that poor job match often happens because people with disability are often placed in jobs purely based on availability. This could be addressed by there being a consistent model that is followed by disability employment consultants to provide a clear and consistent approach and to remove any confusion that employers may have about what support will be provided inside and outside the workplace.

Participants of this study reported that their employment made them feel successful for various reasons including positive mental health, improved independence and having pride in their work. Literature suggests that being employed improves a person's mental health [17] and quality of life as they feel like contributing citizens [20,42]. It is important that employers recognise that a job is not just about productivity and receiving a wage [59], but for many people with ASD employment is important for other valuable psychosocial outcomes, including greater social integration [60]. The opportunity to develop new friendships in the workplace might be particularly important for people with ASD who typically find it challenging to expand their social connections and often experience social isolation [61,62]. Prior research has also shown that the employment of people with ASD can improve workplace culture by allowing for different ways of thinking and performing tasks, as managers are forced to think of more visual and organised ways to present tasks [63].

It was observed that most participants in this study were in entry-level roles with minimal reports of career progression. Career progression is important for everyone as it creates feelings of being valued and of work-life fulfilment [64]. This lack of progression—despite up to eight years in their roles—may be attributed to individuals with ASD feeling content working entry-level jobs as they are just happy to be employed, as suggested in a study by Kopelson [65]. However, for some participants at least, it seems more likely that support is not provided for longer-term career progression and, without that support, people with ASD lack the knowledge and skills to seek higher-level roles. In their paper reflecting critically on what is currently known about the employment of people with ASD, Nicholson and Klag [53] call for more of a focus on long-term success and highlight that more work needs to be conducted on examining what they refer to as the 'employment ecosystem' (demand) rather than focusing extensively on people with ASD and their skills (supply).

Within Australia, this short-term thinking in relation to the employment of people with ASD is likely precipitated by the Australian Government's funding model for the provision of disability employment services [15,66]. Currently, DES providers receive funding if they support a person with a disability to become employed for four, thirteen,

twenty-six and fifty-two weeks [67]. This means that DES providers often focus on support in getting a person into a job and not on job maintenance, as there is no incentive to keep a person in a job past 52 weeks, nor to support their advancement to more senior roles. Many participants were not employed in areas that matched their qualifications nor were many of their qualifications being used in their current workplaces which represents poor investment at the societal level. There needs to be a change in the way funding is provided to DES providers and employers so that there is also a focus on longer-term employment and skills building throughout people with disability's working lives [53].

Despite the funding and practice limitations discussed above, participants in this study indicated they were relatively satisfied with the support they were receiving from their DES provider. However, there were participants who indicated they were not working towards their future career aspirations, but instead in a field that did not align with these aspirations. This could be attributed to the expectation of disability employment consultants that it is better for people with ASD to be employed somewhere, rather than unemployed due to future career aspirations [55]. Despite all participants saying they were happy with the services provided, Nicholas et al. [54] found that people with ASD often found employment services inaccessible and insufficient to meet their needs. In contrast, employment consultants involved in the same study consider the service they provided as being high quality [54]. This shows there is a gap in the expectations of people with ASD and employment consultants. In addition to this, family and parental expectations play a role in the employment of people with ASD, as if parents have low expectations they will not be motivating and encouraging their child with ASD to try to find employment [68]. These low expectations need to be raised so that people with ASD can experience a meaningful career instead of continuing to work an entry-level job with no career progression.

### 4.1. Limitations

The limitations of the study included sample size and diversity of participants which impact the potential for generalisation of results. There were only nine participants included in this study and the sample was male-dominated, which is perhaps not surprising given there is evidence that males are four times more likely to be diagnosed with ASD than females [69]. Despite these limitations, there was evidence of similar lived experiences in other Australian and international studies which gives some confidence in the findings. Though three providers agreed to support recruitment, all participants received support from the same DES provider which also limits the generalisability of the findings as other providers may have different service and support offerings and different levels of expertise in supporting the needs of people with ASD.

### 4.2. Future Research

Whilst this study provides important perspectives, it also highlights opportunities for further research. Future research should focus on looking at a larger sample size and include people from different states in Australia to provide a better understanding of the perspectives of people with ASD around Australia. Future research may also focus on following the different stages of employment for people with ASD. This would provide a better understanding of each stage and the support provided, as some participants in this study struggled to recall the job searching phase in detail. Finally, future research could investigate the perspective of multiple stakeholders such as people with ASD, disability employment consultants and employers. For young adults making the transition from school to the labour market, in particular, it would be useful to also include key family members (such as parents) in career planning and transition. It is evident that family support is important to young people with ASD and this need for family support may extend well into their twenties as evidenced in this study. This additional research would be useful for developing a support framework to promote consistency and meet the needs of employees with ASD as well as their employers. This could be achieved for example

via a Delphi study as a starting point towards the generation of a model to support people with ASD throughout their working lives.

## 5. Conclusions

This study demonstrates that people with ASD feel successful when they are employed, valued, and supported by supervisors and co-workers. However, the study also highlights the impact of underemployment, and the poor return some people with ASD experience on their investment in their own education and skills development. In alignment with existing literature, the study identifies cases of poor job matching, as well as the lack of long-term support for people with ASD to progress in their careers. Notably, disability employment providers are not resourced or incentivised to support longer-term career development and, without it, people with ASD may remain in low-skill, entry-level roles throughout their working lives. There is a need for a consistent model of support which addresses and supports long-term career development needs as well as labour market entry into mainstream open employment.

**Author Contributions:** The study was conceptualised by M.S. with support from C.H. and J.A. Data collection and initial analysis was conducted by M.S. under the supervision of C.H. and J.A. Final analysis and confirmation of themes was undertaken by all authors. The initial draft of the paper was developed by M.S. with critical revisions by C.H. and J.A. All authors have read and agreed to the published version of the manuscript.

**Funding:** This research received no external funding.

**Institutional Review Board Statement:** The study was conducted in accordance with the Declaration of Helsinki and approved by the Social and Behavioral Research Ethics Committee of Flinders University, Adelaide, Australia (Project 8615).

**Informed Consent Statement:** Informed consent was obtained from all subjects involved in the study.

**Data Availability Statement:** The data are not publicly available due to participant consent restrictions.

**Acknowledgments:** We extend our thanks to the disability employment providers who supported recruitment for the project and the nine people with ASD who agreed to give up their time to discuss their employment experiences.

**Conflicts of Interest:** The authors declare no conflict of interest.

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
