# Peer review of "The Lived Experiences and Perspectives of People with Autism Spectrum Disorder in Mainstream Employment in Australia"

_disabilities, doi:10.3390/disabilities2020013_

Round 1
Reviewer 1 Report
Thank you for inviting me to review the paper entitled “The lived experiences and perspectives of people with Autism Spectrum Disorder in mainstream employment in Australia” which aims to examine the support received by individuals with ASD in gaining and maintaining open employment from their perspective.
It is an interesting paper. In line 157, there is a missing word, ‘from’ the same provider? Two main limitations of this article: the small sample size and support from the same provider limit the generalisability of the findings. However, the authors had described these limitations and provided suggestions to overcome these limitations in future research.
The strengths of the paper
- It is a well-written and well-constructed paper. I enjoyed reading it.
Weaknesses
- The small sample size and not much variety in terms of service providers.
- Not sure about the aspects that were covered in the interviews. Authors, please describe or provide sample questions for the semi-structured interviews
Author Response
See attached document for detailed response to each item of feedback

Reviewer 2 Report
It was a pleasure to read this well written article addressing the important topic of autism employment. Overall, the literature review engaged with the relevant and recent literature, however, I was surprised that you did not include the recent report on a global survey of autism employment on the Australian Autism CRC website. Phenomenography is an appropriate methodology to examine the perceptions of autistic adults, in particular, how they understand their employment experience. In order to determine the rigour of the approach you used, it would be helpful to have more detail on how you questioned participants to get at their deep level understanding. Given your use of thematic analysis, I questioned whether phenomenography really was undertaken -- your findings seem to align with a scientific realism or social constructivism epistemology. A phenomenology epistemology would yield different ways of knowing, that is some of your participants had Knowing A, some had Knowing B, etc. This is an easy fix, simply indicating that your epistemology is either scientific realism or social constructivism. I applaud how you connect your findings to policy implications, and think your suggestions are warranted and feasible.
A few minor grammatical issues to address:
Typo at line 116 worth should be work
Typo at line 123 primary should be primarily
Word from missing from line 157
Missing apostrophe in participants at line 169
Who needs to be dropped from line 240
Should be managers not manager at line 314
Author Response

(The authors gave the same response as above.)

Reviewer 3 Report
Thank you for the opportunity to review the manuscript “The lived experiences and perspectives of people with autism spectrum disorder in mainstream employment in Australia”. The authors did an admirable job including rich descriptions of work options, outcomes, and factors contributing to underemployment for adults with ASD, and the authors attempted to advance our knowledge about employment via the perspectives of employed adults. The manuscript is well-written and descriptive, however there are a few issues I would recommend the authors to consider to improve the manuscript.
Introduction
- Language - the authors use person first language, but may consider identity-first language for autistic adults, or maybe at least adding an acknowledgement to these different preferences as preferred by autistic adults (e.g., Kenny et al. 2015, Bury et al. 2020; and see Botha et al. 2021). The Bury et al. (2020) paper might contain considerations that are especially relevant for the current study sample.
- Bury, S. M., Jellett, R., Spoor, J. R., & Hedley, D. (2020). “It Defines Who I Am” or “It’s Something I Have”: What Language Do [Autistic] Australian Adults [on the Autism Spectrum] Prefer? Journal of Autism and Developmental Disorders. https://doi.org/10.1007/s10803-020-04425-3.
- Line 47 - Flint et al is in a different citation format than others
- Line 50 - could the authors clarify what is meant by ‘deliver better mental health’?
- Page 2, paragraph starting on line 52. Authors discuss the benefits of a strengths-based approach, could they give a few more examples or details? Overall, the tone of the introduction thus far has focused on barriers and negative outcomes, so there may be strengths of working adults with autism to highlight in the first paragraph of the manuscript that will help connect to this paragraph and later (the final sentences of the first paragraph in Section 1.1 seems to also reflect this strength-based approach).
- Also on Page 2: With the statement of how “disability employment consultants are not always trained or equipped to support people with ASD…” Lines 58-59, perhaps the authors could add a brief statement of what a disability employment consultant is supposed to do. Is it to emotionally support autisitic adults in job searching, or do they look for jobs for them, etc.?
- Section 1.2: I felt that the introduction was informative, but did not articulate enough that there was a gap in understanding perceptions of adults with ASD. I felt a stronger case could be made in terms of rationale to illustrate how this information would enhance the overall picture of employment for adults.
Methods
- Section 1.2 & Methods - approach - could the authors expand on why a phenomenological approach is being used to understand the study goals? Perhaps expanding upon the first two sentences in the Methods section.
- Line 157: There may be a missing word in this sentence
- Table 1: While I believe this Table to be informative and a helpful amount of detail, I worry about the potential issues of confidentiality, particularly if the recruitment site or employer could identify some of the subjects by these details. Can the authors comment on this aspect of confidentiality (or perhaps provide ranges of ages/length of employment to further anonymise the participants)?
Discussion
- Could the authors reiterate the goals of the current study in the first paragraph of the discussion?
- I felt the implications of family support could be discussed a bit more - are there any recommendations or implications of families supporting the gaining or maintaining of employment?
- Similarly, I felt the point brought up by Participant 1 in the Results, lines 227 - 229, to have a number of implications that the authors could expand upon. Social relationships are hard for autistic adults to come by but they are intricately tied to the experience of well-being and happiness. The statement “I made a friend” is critical.
- The suggestions for future research are specific and connect well with the results of the study.
Author Response

(The authors gave the same response as above.)
